# Global Warming and Its Implications on Nature Tourism at Phinda Private Game Reserve, South Africa

**DOI:** 10.3390/ijerph19095487

**Published:** 2022-04-30

**Authors:** Zinzi E. Sibitane, Kaitano Dube, Limpho Lekaota

**Affiliations:** Department of Tourism and Integrated Communication, Vaal University of Technology 1, Vanderbijlpark 1900, South Africa; zinzisibitane@yahoo.com (Z.E.S.); limphol@vut.ac.za (L.L.)

**Keywords:** extreme temperatures, staff health, heatstroke, sleep disturbance, climate change

## Abstract

The past decade recorded the highest number of high impact extreme weather events such as flooding, rainfall events, fires, droughts, and heatwaves amongst others. One of the key features and drivers of extreme weather events has been global warming, with record temperatures recorded globally. The World Meteorological Organization indicated that the 2010–2020 decade was one of the warmest on record. Continued global warming triggers a chain of positive feedback with far-reaching adverse implications on the environment and socio-economic activities. The tourism industry fears that increased global warming would result in severe challenges for the sector. The challenges include species extinction, disruption of tourism aviation, and several tourism activities. Given the extent of climate variability and change, this study examines the impacts of rising temperatures on tourism operations at Phinda Private Game Reserve in South Africa. The study adopts a mixed-method approach that uses secondary, archival, and primary data collected through interviews and field observations to investigate the impacts. Data analysis was done using XLSTAT and Mann–Kendall Trend Analysis to analyse climate trends, while content and thematic analyses were used to analyse primary data findings. The study found that increasing temperature is challenging for tourists and tourism employees as it affects productivity, sleeping patterns, tourism operations, and infrastructure. High temperatures are a considerable threat to water availability and animal sightings, adversely affecting the game drive experience. Increased heatwaves resulted in bird mortality and hatching mortality for turtles; this is a significant conservation challenge. The study recommends that heat stress be treated as a health and safety issue to protect tourists and employees.

## 1. Background and Introduction

One of the critical aspects raised prominently in the Paris Agreement is the need to curb the temperature between 1.5 °C and 2 °C to preindustrial levels to avoid disruptive climate events buoyed by global warming. Tipping beyond the 2 °C threshold is widely believed to have catastrophic implications for human civilisation as we know it [1,2]. The Special Report on Global Warming of 1.5 °C (SR15), which was published in 2018 by the Intergovernmental Panel on Climate Change (IPCC), paints a gloomy picture of the future if the world fails to drastically reduce its carbon emissions responsible for global warming [3]. There are fears that a continued increase in global warming would adversely affect the achievement of the Sustainable Development Goals. Regardless of the devastating implications of continued anthropogenic warming, evidence suggests that some parts of the world are already experiencing temperatures above the 1.5 °C and 2 °C temperature threshold, with some scientists noting that the world has already entered a climate emergency [4,5].

According to the World Meteorological Organization, 2016 and 2020 emerged as two of the hottest years on record since the documentation of climate data [6]. A global average increase in temperature is a huge concern given that it has adverse impacts on international climatic elements, which has inevitable consequences on human civilisation and various tourism activities.

Global temperature drives global weather patterns. There is clear evidence that changes in temperature result in changes to global climate patterns with far-reaching adverse impacts on economies and society [7,8]. Various regions are affected differently by increased temperature, given different vulnerabilities, resilience, and adaptation capacity. An increase in temperature has also been blamed for the increased frequency of fires and fire-related damages and costs [9,10,11]. Fire risk has been of particular concern to tourism stakeholders as it threatens tourism properties, employees, and tourists [12]. Dube [13] noted temperature increases as one of the drivers of many destructive fires threatening many nature destinations in the southern African region, calling for proactive measures to address this growing concern.

Regarding tourism, one of the main worries in the northern hemisphere has been the impact of temperature increases on winter tourism and tourism activities such as skiing [8]. According to Damm et al. [14], under the 2 °C scenario, Europe’s ski tourism industry is expected to experience losses in demand, with the most affected countries being Austria and Italy. According to Steiger and Scott [15], in other destinations losses in the length of ski seasons are a significant worry with declines of as much as 16% in seasonal length. To offset the impact, companies reliant on winter snow in both the northern and southern hemispheres have increased efforts to invest in snowmaking to reduce the sector’s sensitivity and vulnerability to global average temperature increase [16,17,18]. However, adopting such measures equally comes at a substantial economic cost and could drive the cost of tourism business in the polar regions. There is also a concern that global warming will adversely affect the Winter Olympics. A study by Scott et al. [19] (p. 913) noted that the Winter Olympics are under threat as evidence points to a decline of “probability of a minimum temperature of ≤0 °C”, which is not ideal for such an event.

The Mediterranean region, whose main attraction is beach tourism, is set to be adversely affected by an increase in heatwaves, pushing tourists to other areas with cooler temperatures [8]. An increase in global temperature in the Australian region is expected to worsen coral bleaching in the Great Barrier Reef [20,21] and threaten nature tourism [22,23]. This has given rise to fears around last chance tourism in the region which has been a considerable concern for the United Nations Educational, Scientific and Cultural Organization (UNESCO) that oversees world heritage sites.

Africa is one of the regions already experiencing a climate backlash due to global warming. As the amount of carbon emissions increases, the impacts of global warming will become more pronounced on people and nature. Nature tourism is one of the economic sectors feared to suffer some of the worst impacts of global warming on the continent [24]. The State of the Climate in Africa 2020 noted that the surface temperature in Africa increased on average between 0.45 °C and 0.86 °C from 1981–2010 [25]. The IPCC Sixth Assessment Report (AR6) noted that hot extremes across Africa have emerged above the global average, using the period between 1850 and 1900 as a reference point [26]. This will worsen the melting of glaciers at mountain tourist destinations across Africa such as Mount Kenya, the Rwenzori Mountains in Uganda, and Kilimanjaro in Kenya [27,28,29,30].

Temperature rise is also expected to result in adverse modifications to coastal tourism destinations on the continent. Global warming results in increased coastal inundation and flooding due to the thermal expansion of oceans caused by global warming, adversely affecting beaches and tourism properties along the coastline and implications for tourists’ experiences and activities [31,32]. Some regional hotspots for coastal tourism damage due to global warming include Mombasa, Kenya [33,34], Egypt [35], Seychelles [36], Morocco [37], and Ghana [38]. Additionally, beaches threatened with an inundation of increasing temperature adversely impacts corals, which are the mainstay of snorkelling activities [39,40].

In southern Africa, concerns have been rife about the implications of rising temperatures on the comfort of tourists in some destinations. Dube et al. [41] and Dube and Nhamo [42] show that tourists and tour operators in Victoria Falls are worried about extreme temperatures. It adversely affects outdoor tourism and recreational activities, with implications for tourists’ itineraries. The situation is likely to worsen, as the region is warming faster than the world average according to a study by Lennard et al. [43].

While there is some progress regarding progress in tourism and climate change knowledge, there are still many uncertainties about how various climate parameters will affect the sector and its activities across the world and, in this case, Africa in particular. According to Scott et al. [44], although tourism and climate studies increased in the Fifth Assessment Report (AR5), there are still considerable regional knowledge gaps on tourism and climate change, making it challenging to adopt concrete resilient measures for the sector. Africa is one such area where this vast knowledge gap is prevalent. Despite some fragmented studies in response to tourism and climate change in recent times, these have not been sufficient to cover all geographic areas [44]. Therefore, this study originates from the desire to reduce this scientific gap. This study aims to assess the impacts of increasing temperature on tourism, particularly nature-based tourism, at private game reserves in South Africa, using Phinda Private Game Reserve as a case study.

There are two critical questions that this study seeks to answer: (i) What are the temperature trends at Phinda Private Game Reserve? (ii) How does the increasing temperature in the context of global warming affect nature-based tourism operations in South African private game reserves? This study examines how nature-based tourism is affected by global warming and extreme temperature events such as the increased episodes of heatwaves in the region. Regardless of such vulnerabilities, or evidence of potential exposures to global warming, limited dedicated studies have been conducted to ascertain the impact of rising temperature as a climate parameter on tourism in various African destinations. Therefore, this study explores the effects of increasing temperatures and extremely hot days in the context of global warming on nature-based tourism, focusing on employees, tourist activities, and flora and fauna as they pertain to tourism at Phinda Private Game Reserve, South Africa.

## 2. Study Area

The study was conducted at Phinda Private Game Reserve (Figure 1). Phinda Private Game reserve is one of the private safaris and game reserves owned and managed by andBeyond, one of the world’s leading luxury experiential travel companies. This site is located on a 28,555 hectare (70,560 acre) property on the fringes of iSimangaliso (St Lucia) Wetland Park, a World Heritage Site situated in KwaZulu-Natal Province. It is home to the Big Five (lion, leopard, elephant, buffalo, and rhino) and boasts many other animal species such as cheetahs and 436 bird species [45]. About 95% of the reserve lies beneath 100 meters above sea level [46]. The reserve is unique in that it comprises 1000 hectares (2471 acres) of Africa’s remaining rare dry sand forest. The game reserve is an important conservation area and features six unique luxury lodges. The lodges at Phinda have been nominated for some of the most globally prestigious tourism awards. Other essential features include a private landing strip.

Phinda receives an approximate average of 1000 mm of rainfall annually [45], with rains expected at any time. Rainfall is typically short-lived and permeates in the form of short cloud burst thunderstorms. The area is generally warm and humid, with the Indian Ocean playing a controlling factor. Summer temperatures can reach very high levels, with a daily temperature range between 25 °C and 35 °C [45]. The greater area of St Lucia has been exposed to impacts of tropical cyclones and droughts in the recent past.

## 3. Research Methodology

A mixed-method approach was implemented for this research, requiring various data types to respond to the research questions. To track the temperature regime for the area, the study utilised archival data from South African Weather Services, the principal organisation mandated to provide climate services information in South Africa. Data from Mbazwana Airfield Meteorological station was also used as it is the only station with a longer climate record that could be reasonably analysed. The station is about 50 km from Phinda Private Game Reserve. The airport station is 82 m above sea level, while the altitude of Phinda ranges from 48 m to about 112 m depending on location. A climate time series analysis was conducted using XLSTAT 2021 to run Mann–Kendall Trend Tests on climate data to understand climatic trends. A Mann–Kendall Trend Test is performed to statistically test a monotonic upward or downward trend over a given period. It is a statistically accepted method used in the tourism geography field to assess climatic trends and their significance [47,48]. For this study, a default confidence interval was set at 95%.

Primary data were collected from staff that work at the game reserve and the head office in Johannesburg. These include employees from all sections of the game reserve, ranging from senior management such as hospitality and tourism managers, lodge managers, and conservations managers to kitchen staff, game rangers, tour guides, operations staff, and community leaders. The study sought and gathered evidence from game reserve employees, whose ages range between the ages of 25 years and 65 years. The study used a non-probability sampling method to select the participants. Only employees that have worked in the game reserve for about five years and above were considered for interviews. The period was deemed to be adequate to have witnessed some form of weather and climate events to form an informed opinion on climatic events in the area. Both convenience and snowball sampling techniques were used to identify research participants, with the job description and climate exposure being the main selection criteria to determine participants. A total of 30 in-depth interviews were conducted with staff.

Interviews were conducted in May 2021. A semi-structured interview guide was used. During the interviews, probing was used to solicit additional information. Depending on the staff member’s seniority, the discussions took approximately 30 min to an hour. All interviews were conducted at Phinda Game Reserve, mainly at Mountain Lodge and Forest Lodge. The interviews aimed to gather data on how increases in temperature and heatwaves affect tourism operations, guests, employees, flora, fauna, and infrastructure at the game reserve. The perceptions concerning temperature trends, prospects, and other climatic aspects of the game reserve were also examined. During the interviews, notes were recorded. Participation was through informed consent.

Post interviews, the audio recordings were transcribed into a word document. After the data were transcribed, reading of the data was undertaken to familiarise with the data and obtain a basic data pattern. The second step was to relook at the research questions to see which questions could be answered using the collected data. To establish a framework, a process of identifying broader concepts, ideas, and phrases was undertaken. This resulted in code generation through the structuring and labelling of data. The data categories and their subcategories and themes were based on the research questions and sub-research questions. The last process entailed identifying data patterns and connections as part of theme development and areas for further exploration. This process followed guidelines set out by Vaismoradi et al. [49]. Figure 2 shows the key steps followed in the data analysis process. One of the key advantages of content analysis is that it can allow quantification and further data analysis. Content analysis also allows the researcher to look at and characterise the data to determine who will have “said what to whom and with what effect,” as noted by Bloor and Wood [50]. Content and thematic analyses provided the researcher leeway for deductive and inductive meanings of the interview contents.

The study was conducted following the approved Ethics Standards of the parent institution. The study had ethics approval number FREC/HS/02/10/2020/6.1.2.

## 4. Results and Discussion

### 4.1. Climatic Trends at Phinda Game Reserve

The study found that the annual average maximum temperature in the area had increased throughout the study period (Figure 3a). On the other hand, the average yearly minimum temperature decreased over the same study period, indicating an increased variance between the annual average maximum and minimum temperatures. Over the period, the average yearly maximum temperature had grown from 26.5 °C to about 27.3 °C, signifying an annual average temperature increase of about 0.8 °C over 27 years. The temperature for the area was determined to be about 26.9 °C with a standard deviation of 0.97. Given that the computed *p*-value was 0.190, greater than the significant alpha of 0.05, the temperature increase was statistically not significant. On the contrary, a statistically significant decline in annual average temperature decreased as it declined from about 17.5 °C to about 16.3 °C, marking a drop of about 1.2 °C (Figure 3b). The *p*-value was found to be 0.009, lower than the significant alpha of 0.05.

The study found that there was not a statistically significant increase in annual average maximum temperature. On the other hand, some months reported a statistically significant increase in temperature. To be precise, four months had witnessed statistically significant maximum temperature increases, namely February, May, June and July, as indicated in Table 1. Generally, all months showed that maximum temperatures were increasing, although this increase was not statistically significant. In summer, only one month (February) witnessed a statistically significant increase in temperature, while the winter months (June and July) showed signs of significant warming. May recorded the highest significant temperature increase, where the temperatures pushed into the mid-20s.

On the other hand, monthly minimum temperatures for several months recorded a decline, except for March, April, and May, where there were ineligible temperature changes. The rest of the months recorded significant mean monthly temperature declines, as indicated in Table 2. This result was unexpected as there was an anticipation that the temperature would generally increase in line with the average global and regional temperature trajectory [26]. This might signify an increase in the daily diurnal temperature range. Even though the average minimum temperatures declined over the study period, there is no evidence of minimal extremes and sub-zero temperatures. The area seemingly had high minimal temperatures ranging in the 20s.

The area where Phinda is located is experiencing a statistically significant increase in the number of hot days whose temperatures are between 32 °C and 35 °C, as indicated in Figure 4. There is a general increase in the number of extremely hot days in temperatures above 36 °C to 39 °C and above 40 °C. Such temperatures are warmer than the region’s average temperature, which is set at a mean annual average of between 22 °C and 24 °C [51]. The results confirm that the area is one of the warmest in the country, as already reported [51]. Such high temperatures pose a health risk to tourists and tourism employees, particularly those with underlying conditions and the elderly. This is in line with the anticipated increase in hot days globally [3]. The study found that the number of sweltering days (above 40 °C) is concentrated between October, November, December, January, and April. The highest number of extremely hot days was reported in December.

While the increase in average maximum temperature was expected, the decline in monthly minimum temperature was an unexpected occurrence as, ordinarily, the observation is that temperature trends are on the increase globally [30]. The increase in monthly and daily temperature and a decline in minimum temperature means that the daily temperature ranges for the area can be quite high on some days, which could have various impacts on human physiology and temperament. The climatic data was confirmed by one of the respondents who noted that: “… this area is very hot and, on some days, the temperature can reach as high as 48 °C…”.

A high daily temperature range impacts energy use at the Phinda Private Game Reserve, especially at the lodges and camps, as it means an increase in demand for energy for cooling and heating purposes. Energy demands would increase during those days when it is extremely hot for cooling purposes, and air conditioners would be needed to warm up the places during the night, early hours, and or days when the temperature drops drastically. During the day, temperatures tend to be very high, with some day temperatures surpassing the 40 °C temperature mark (Figure 4). The demand for cooling will be high, while on the other hand, extremely low temperatures will escalate the need for heating using air conditioners. There is also an anticipated demand for heating of water by guests using geysers. Increased air conditioners will inevitably lead to more energy use and more carbon emissions, creating a spiral of global warming since the lodge is not carbon neutral yet and is heavily dependent on energy from the national grid.

The observed warming in some winter months resonates well with other findings from elsewhere in the world. The warming of the winter months is threatening tourist activities such as the Winter Olympics in the global north [28]. In Southern Africa, the warming of winters is expected to affect the sustainability of snow at Afriski Mountain Resort in Lesotho, with consequences for tourism revenue. For game reserves such as Phinda Private Game Reserve, some animal species that thrive in winter can be affected with long-lasting impacts. This assertion resonates well with observations by Marshall et al. [52], who noted that an increase in winter temperature has lasting effects on insects and spring phenology in a complex way; they requested research in determining the net losers and winners of such a development given that the game reserve has diverse flora and fauna that form part of its tourism offering.

### 4.2. Employee Perceptions on the Impact of Global Warming on Tourism

The impact of global warming, particularly extreme temperatures, is particularly concerning for employees at Phinda, who were the main focus of this study. As much as there was an acknowledgement that the area is generally warm, the employees complained that the temperature was notably higher in Hluhluwe. The area’s high humidity also worsened the temperature, given its proximity to the Indian Ocean (a warm ocean) and the St Lucia Wetlands Park and other rivers, which act as humidity sources. The impacts of warming in the area profoundly impact employees’ performance, with many complaining of heat fatigue on certain days. Community members from host communities also complained of battling to sleep on hot days due to excessive heat at night.

Of the employees who participated in the study, 75% indicated that they sometimes experienced difficulty sleeping at night due to the heat. The most affected employees were those with no access to air-conditioned rooms. The employees, who used fans on hot days, were not entirely happy. They noted that on days where the heat is excessive, there is a tendency for fans to recycle and blow off hot air, which defeats the purpose of switching them on. Disturbed sleep due to heat was of particular concern to tour guides, given their work schedule and the fact that they work outdoors. Respondent XV indicated that “on very hot days, we battle to sleep. Given that we have to work long hours and wake up early in time for the morning drive, it becomes particularly challenging as we are always tired”.

Disturbed sleep is particularly challenging as it threatens employee performance and alertness in a demanding sector. Studies conducted in other parts of the world indicate that lack of sleep due to global warming also increases with dire consequences for the people. Matsui [53] noted that warmer nights were defined as any night temperatures above 25 °C. Sleep is a physiological requirement, but lack of sleep is attributed to increased illness, particularly among people suffering from heart diseases. Obradovich et al. [54] observed that lack of sleep was prevalent amongst the elderly, particularly the marginalised. Women were seen to suffer the worst effects due to lack of sleep. Consequently, lack of sleep also increased people’s susceptibility to diseases and chronic sickness and adversely impacted human beings’ psychological and cognitive functioning. Therefore, there is a need to ensure the health and protection of tourists and tourism employees in such environments, as disturbed sleep tends to be a common feature in the tourism sector within the southern Africa region [55].

Regarding the operational side of things, high temperatures were reportedly challenging for staff working in the fields, including operations teams and rangers. The interviewed rangers indicated that their vehicles do not have roof coverings, which presented challenges, particularly on hot days as they battle the direct sunlight heat. The tour guides make concerted efforts to protect themselves against the heat by wearing hats and long shirts. Almost all the tour guides indicated that, unfortunately, this does not make much of a difference when temperatures are very high. They also indicated that tourists are temperamental on extremely hot days when they go out on game drives and often show irritability when tour guides stop for commonly sighted species. Respondent XS noted that “It is often challenging as tourists want to quickly go through the game drives and return to their rooms where it is cooler”.

The effect of heat on tourists was also observed by hospitality staff who work at the lodges. These employees advised that tourists spend time in their rooms and the swimming pool to escape the heat on scorching days. The high temperatures also affect the game viewing experience for the tourists as some animals, particularly the big cats, hide under trees and amongst the bushes due to the heat. Given that animals hide in trees to avoid the heat during the day, this adversely affects game viewing and tourist satisfaction as they do not get to see as many animals as they would like to see.

The kitchen staff also complained that extremely hot days present challenges for their work as it becomes unbearably hot. On hot days, the heat extraction fans that ordinarily function as heat regulators cannot reduce the heat levels in the kitchens, making it difficult to work. Kitchen spaces might require an additional cooling system to ensure that they are conducive to the working staff. In addition, staff working in the massage spa also indicated that hot days are not conducive for massage treatments. It becomes uncomfortable to work, because people tend to be sweaty. In an effort to resolve this challenge, their spa temperatures are airconditioned at 22 °C, a temperature that was determined to be ideal for massage.

The maintenance team that works at the lodges indicated that they have also been faced with an overrunning of air conditioners on hot days, resulting in more equipment malfunctioning. They also reported that the demand for air conditioning maintenance and gas refill increases, attributed to increased usage. There were suspicions and complaints about fridges and air conditioners not working correctly during very hot days. This calls for gadget manufacturers to revise their product manufacturing to cater for increased temperatures. In some cases, the staff resorted to pre-cooling the rooms before tourists arrive so that when the tourists check in, the rooms are at the right temperature. This pushes up energy costs and the carbon footprint of tourism operations. Encompassing green building design can be an alternative way to deal with these challenges.

The study confirms the recent findings by the IPCC [56], which noted an increase in the number of people being affected by heatstroke, with severe cases resulting in death. Sleep disturbances amongst employees will reduce their productivity. However, there might also be a need to take extra care of tourists who might not be acclimatised to the climate to protect them from excessive heat stress. In particular, the management of adult tourists and other vulnerable groups from the global north has to be closely managed to reduce the risk associated with heat stress and stroke within the reserve and other areas. During certain times of the year, heat stress has to be treated as a health and safety issue that the reserve needs to plan for and manage. There is a need and demand to ensure that cooling is taken as a central element of employee building design and construction to ensure that staff welfare is taken care of.

The Information and Technology (IT) team complained that extremely hot days were problematic for IT infrastructure. They often damage critical communication infrastructure, which they have to repair frequently when it becomes very hot. SFPs (small form-factor pluggable), servers, and bit switches were particularly vulnerable to excessive heat. In particular, the SFP at the Mountain Lodge was reportedly getting burnt more often than other areas when temperatures were very high. The increased need for maintenance and IT maintenance escalated operational costs. Communication was also disrupted by such occurrences in and around the lodge, which is a challenge. The SFP temperature sensitivity and its effect on functionality are well documented by Kodet et al. [57]. Given the demand for connectivity [58], there are chances that disrupted communication can affect tourist satisfaction and experience in such instances.

### 4.3. Impact of Global Warming on Animals

Besides the impact of temperature on human and tourism infrastructure, the rising temperature was observed to hurt bird species in the game reserve. Employees indicated “...that most birds populate near roofs on very hot days and, in some cases, fall and die”.

The vulnerability of birds to excessive temperature is documented, with studies indicating that birds struggle to survive temperatures over the mid-30s [58]. According to Moagi et al. [59], when temperatures are between 35 °C and 40 °C, birds reduce foraging efficiency, body weight, and breeding success. The study also revealed that sweltering and extremely hot days have increased over the years. This is a disturbing trend as Phinda considers birding one of its unique tourism offerings. The continued death of birds can negatively affect bird sightings and tourist satisfaction for birders. The death of birds observed at Phinda Private Game Reserve raises the alarm about the future of biodiversity in the African region, which is also battling poaching challenges and overharvesting. Some fear that increased heat stress and temperature overshoot will negatively affect bird populations and drastically reduce their population in southern Africa [60]. The death of birds has severe ecosystem implications. This raises the demand for global citizens to strengthen efforts to drive down carbon emissions to avert the challenges associated with temperature overshooting.

During the interviews, one of the issues that arose was that tourists who visited the game reserve enjoyed the trip to Sodwana Bay along the coast. Besides swimming, one of the popular activities enjoyed by tourists is watching turtles. The area serves as a breeding ground for turtles. Increasing temperatures are a threat to the turtles. One conservationist indicated that an increase in temperature due to global warming tends to distort turtle birth sex ratios, which could alter the population growth of these turtles. This observation is supported by Esteban et al. [61], who reported that an increase in sand or beach temperature where turtles hatch tends to skew the sex ratio to more females than males. As temperatures increase due to global warming, the hatching mortality will also increase, which can threaten the turtle population. There are also concerns amongst ecologists over the decreasing body size of turtles due to global warming, which are believed to be signs of a lack of resilience amongst turtles [62]. Such developments require strategic conservation efforts to protect the turtle population, affecting tourist activities. The increasing temperature was also a worry as it affected water availability. The game reserve has several small earth dams that are not deep. An increase in temperature, especially during drought years, will increase the potential of such small dams drying due to increased evaporation rates, as seen in other protected areas [63]. Employees reported that with increased droughts, they had faced water challenges in the game reserve that led to the closure of one of the camps utilised by the staff. Such small dams and weirs are critical for biodiversity protection and aquatic life. They also act as a bird sanctuary as they escape heat waves [64].

## 5. Conclusions

The study sought to examine the trend and impacts of extremely high temperatures in the context of global warming at Phinda Private Game Reserve. The study found that the number of hot and extremely hot days is on the increase at Phinda Private Game Reserve, which contributes to a severe impact on tourism operations at the reserve. There is also a significant increase in the number of extremely hot days and heatwave days. The results resonate well with the findings of the AR6 that reported that the number of heatwaves and extremely hot days would increase in that region. The high number of days with temperatures above 40 °C is concerning from a health and safety perspective as it has implications for flora and fauna. The study found that these extremely high temperature and heatwave days are a cause for concern as they adversely impact the reserve’s employees, tourism operations, and animals.

The employees reported that they battle to sleep at night on extremely hot days and battle to concentrate and work during the day. This, in turn, affected productivity at work, more so for those low-end staff without access to air conditioners. Staff also indicated that this equally affected their morale as it caused stress. High temperatures during the day particularly affected field staff such as game rangers, tour guides, and grounds staff. The heat seriously impaired the capacity of staff to perform at their best. Tourists were reportedly temperamental on very hot days, and animal sightings were adversely affected as animals escaped the heat under the shade of trees, making sightings difficult.

The excessive heat also increased evaporation rates, adversely affecting water availability, particularly in drought years, with adverse implications for employees, tourists, and animals. Bird species were also dying due to heatwaves, threatening the ecosystem and biodiversity. The turtles in Sodwana Bay were also reportedly under threat due to high temperatures that affected the hatching rates and gender of the turtles.

Therefore, the study recommends that to protect rangers and tourists on such days, there is a need to: provide vehicles with sun covering to ensure health, safety, and comfort, particularly for the old and vulnerable, provide adequate cooling for staff and green building investment aimed at reducing carbon emissions, and to address the issue of energy efficiency and climate adaptation through appropriate climate insulation of buildings. Retrofits can be a remedy where new green buildings cannot be established. Staff accommodation must also be adapted to the region’s present and future climate scenarios. Some activities can also be adjusted to consider the daily high temperatures to reduce the risk of sunburns, heatstroke, and other health challenges associated with environments with very high temperatures. The issue of the carbon risk to the tourism industry cannot be decoupled from the issue of climate change, as such reasonable steps have to be taken to ensure that the company either offsets its carbon footprint or develops mechanisms for tourists to purchase carbon credits to offset their carbon footprint. The company can also employ electric vehicles whenever this is feasible to cut back on the carbon footprint. Future studies can look at how tourists have been able to withstand and survive such heat and their take on some of the critical issues raised in this study. There is an urgent demand for monitoring how species respond to such temperatures and adopting conservation measures to ensure the protection of vulnerable species.

### Study Limitation

The unavailability of long-term data is a challenge to the confidence levels of the study and its findings.

## Figures and Tables

**Figure 1 ijerph-19-05487-f001:**
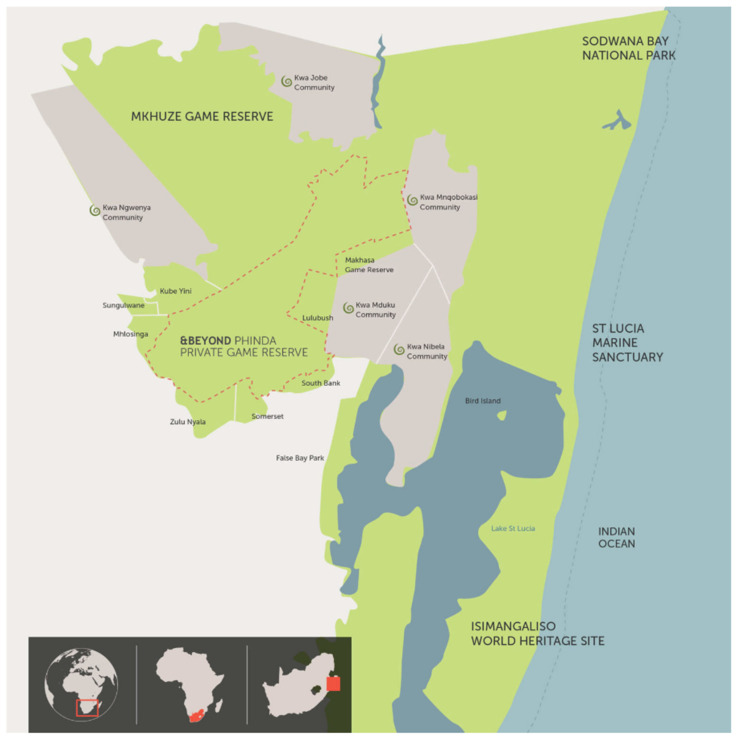
Phinda Private Game Reserve, KwaZulu-Natal Province, South Africa. Source: Image Supplied by andBeyond.

**Figure 2 ijerph-19-05487-f002:**
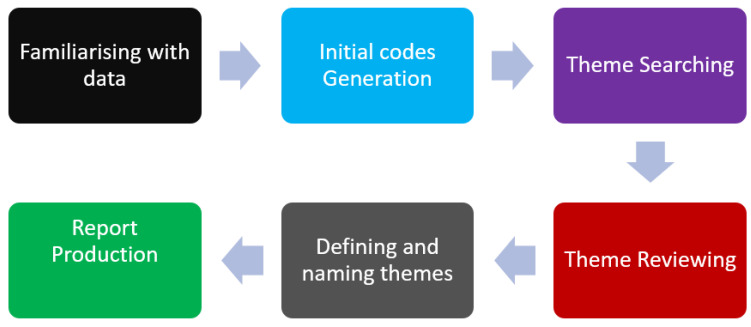
Steps and procedures followed during content analysis. Source: Authors.

**Figure 3 ijerph-19-05487-f003:**
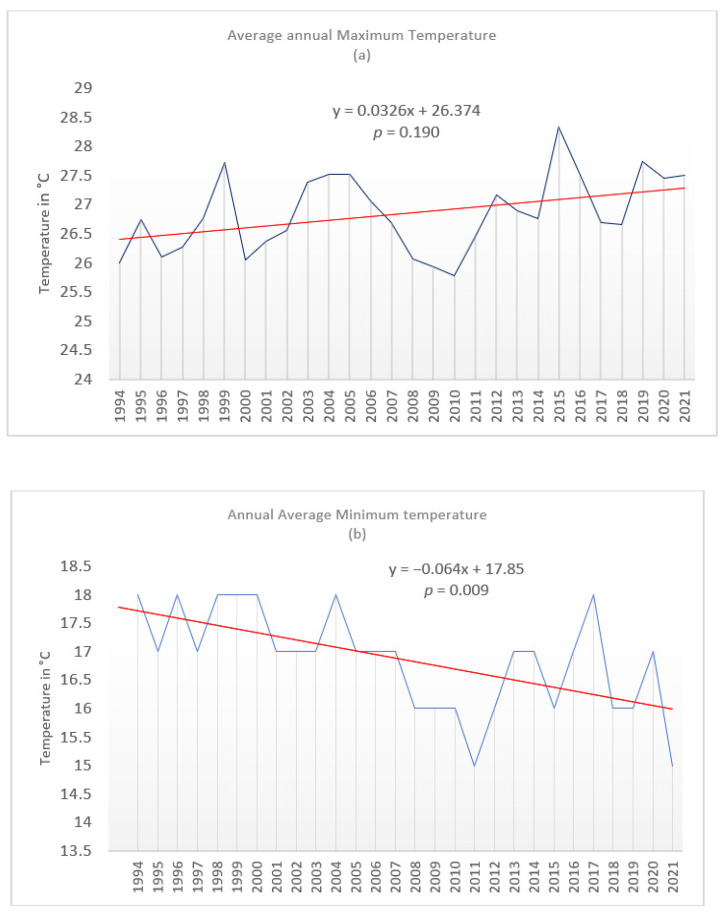
Annual average maximum and minimum temperatures for Mbazwana 1994–2021. Source: Authors.

**Figure 4 ijerph-19-05487-f004:**
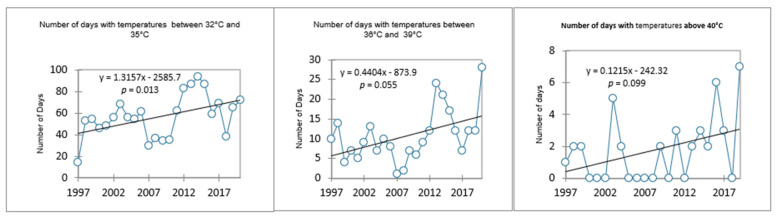
The trend in the number of hot and extremely hot days in Hluhluwe.

**Table 1 ijerph-19-05487-t001:** Months with significant temperature increase °C (significant alpha is 0.05).

Month	*p*-Value	Temperature in 1994	Temperature in 2021	Temperature Increase	Monthly Average Temperature	**Standard Deviation**
**February**	0.049	29	30.2	1.2	29.8	1.167
**May**	0.007	25	27	2	25.9	1.074
**June**	0.038	23.9	25.1	1.2	24.5	0.820
**July**	0.012	23.8	24.9	1.1	23.7	1.285

**Table 2 ijerph-19-05487-t002:** Months with significant temperature decrease in °C.

Month	*p*-Value	Temperature in 1994	Temperature in 2021	Temperature Decline	Long Term Monthly Average Temperature	Standard Deviation
**January**	0.002	22	20	−2	21	1.071
**February**	0.024	21.5	20	−1.5	20.8	1.086
**September**	0.004	16.2	14.2	−2	15.5	1.209
**October**	0.003	17	16	−1	17.1	1.269
**November**	0.007	20	18	−2	18.8	1.116

## Data Availability

Data available on request.

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
