# Peer review of "Global Warming and Its Implications on Nature Tourism at Phinda Private Game Reserve, South Africa"

_ijerph, 2022, doi:10.3390/ijerph19095487_

Round 1

Reviewer 1 Report

Dear Authors,

I really enjoyed in reading your manuscript which is well presented as well as well-structured in any part.

The research question “The primary question is, how does the increasing temperature in the context of global warming affect nature-based tourism operations in South African private game reserves” is clearly stated since the beginning of the paper and it is also developed both in Results and Discussion of Results accordingly. Moreover, the object of research has been well specified. In my view, your paper is almost ready to be published.

I just provide Authors some suggestions in order to improve the quality of their paper:

- Since the Abstract please specify the nature of your research which is clearly qualitative (lines 19-20) and then in Material and Methods

- In Literature please clarify “AR6” (p. 3, line 105) and have a look at: “The Over the past decade” (line 78)

- In Materials and Methods, Authors state that “Regardless of such vulnerabilities, or evidence of potential vulnerabilities to global warming, limited dedicated studies have been conducted to ascertain the impact of rising temperature as a climate parameter on tourism in various African destinations. Therefore, this study explores the impacts of increasing temperatures and extremely hot days in the context of global warming on nature-based tourism, focusing on employees and nature-based tourism activities at South African game reserves. The study also examines the impacts of global warming, particularly the increasing temperatures on flora and fauna, at Phinda Private Game Reserve in South Africa.” Therefore, I assume that they explore two research questions and not “multiple research questions” as they wrote in line 159.

- Another point that I suggest you to improve, regards the results of the content analysis: frame a figure in Results in order to show how you implemented the scheme outlined by  Fig. 2. I think you can add more value to your work by linking your theoretical representation of the content analysis  to the content analysis results.

I wish you good luck for your work

Author Response

Response to reviewer attached 

Reviewer 2 Report

A brief summary

This paper addresses a relationship that I consider important: global warming and nature tourism. It uses historical climate records, specifically data over 24-26 years. However, according to experts, the ideal is to have time series of at least 30 years to detect trends in the climate. It also uses interviews from staff that works at the game reserve and the head office in Johannesburg. The obtained results are expected, but no less interesting for that, and the authors relate them appropriately with possible future scenarios for the nature tourism at Phinda Private Game Reserve. Its reading is generally easy. The wording is clear, except for some problem related to the recurring use of certain expressions or words in sentences that are too close together (this is commented after). The use of English is also correct. However, the work presents a series of problems that must be resolved before proceeding with its publication. They are fundamentally related to the structure of the article, which has some serious flaws.

Sections 1 and 2 (Introduction and Literature) should be merged to create only one (Introduction). There are messages, problems and structures that are repeated in both sections and seem like two versions of the same text. I am not proposing merge them simply by removing the section title (line 77). The text should be revised to combine both sections, since they advance in parallel through the same topics.

The same happens with the Results and the Discussion sections. They are now separate and should be together (Results and Discussion). Why? Because in the current results, the authors already make a first discussion of many of the outcomes. In addition, the current discussion section is very limited, with just 20 lines and even repeating some arguments that were already provided in the Results section. Again, the solution is merge these two sections (currently 4 and 5) combining the current texts and deleting repetitions.

There are also some mistakes in the citation of references in the main text. The number of definitive references will be higher once these missing citations will be added (I specify them later).

The climatic data shown in figures 3 and 4 do not correspond to the same time periods. This is inelegant, but not a mistake. However, in Figure 3 the authors should make some more work. One graph goes from 1994 to 2019 and the other one from 1995 to 2020. This does not coincide with what is expressed in the caption. Please modify the first graph, or correct the caption to add the other period.

Finally, I detail a few minor problems in the next section.

Specific Comments

Citation: I do not know why you use names in citation for the two first authors instead their surnames.

Lines 5 and 7 (Affiliations): use capitals for “integrated” in both affiliations.

Line 11 (Abstract): delete “such events”. You are using this expression in the previous line (Line 10). Use “others” instead “other” to be grammatically correct.

Line 28 (Keywords): search engines use words present in the title, abstract and this section. If you are repeating words used in the title at the keywords, you are losing visibility. I suggest change four of the five current keywords: global warming, nature tourism, game reserves, South Africa. You could use for example: extreme temperatures, staff health, heatstroke, sleep disturbance.

Lines 53, 63, 69-70 (Introduction): avoid repeat the same expression. “worst affected regions affected by”; “some progress regarding progress in tourism”; “knowledge gap”, which is repeated in lines 69, 70 and 72.

Line 66 (Introduction): Scott et al. [18] instead of showing all the co-authors.

Line 76 (Introduction): delete the quote after the question mark.

Line 89 (Literature): delete spaces after numbers in the reference ([20; 21; 22]).

Line 104 (Literature): delete the reference, because it was already showed in line 101.

Lines 108-112 (Literature): this text could be shortened. Summarize and do not repeat the same idea.

Lines 123-124 (Literature): quote correctly. This alter the references numbering. Do not forget checking it.

Line 126 (Literature): again, use Lennard et al. instead of showing all the co-authors.

Line 138 (Material and Methods): create a sub-section named “Study area” for text between lines 138 and 157.

Lines 138-139 (Material and Methods): “This site” instead of “Phinda Private Game Reserve”, since this text is in the previous sentence.

Figure 1: for the caption, use the complete text provided by García et al. (2015). “Light grey polygons show state-run protected areas. The white background is a mosaic of agricultural and communal lands and the light blue polygon shows the Lake St-Lucia. The inset on the left shows the location of Phinda in South Africa.” Add the reference in the list of your paper: Rostro-García, S., Kamler, J. F., & Hunter, L. T. B. (2015). To Kill, Stay or Flee: The Effects of Lions and Landscape Factors on Habitat and Kill Site Selection of Cheetahs in South Africa. PLoS ONE, 10(2), e0117743. https://doi.org/10.1371/journal.pone.0117743

Line 158 (Material and Methods): create a sub-section named “Climate data recording” for text between lines 158 and 171.

Line 172 (Material and Methods): create a sub-section named “Interviews” for text between lines 172 and 203.

Line 177 (Material and Methods): “convenience” instead of “convenient”.

Line 180 (Material and Methods): you are talking about a semi-structured interview. Describe the aims of the interview and the questions addressed.

Lines 205-207 (Results): delete this text, is a journal’s guideline for address this section.

Lines 212-220: use always “annual average maximum temperature” and “annual average minimum temperature” along this text. Be careful, because the order of words is different in line 212, and in other cases, some words are absent (annual, maximum, minimum). Be always consistent. Delete “annual average temperature” in lines 213-214, is not necessary to understand the data.

Lines 217 and 224 (Results): “non-significant” instead of “insignificant”, because both concepts are different in statistics (in fact, the latter does not exists).

Lines 257-258 (Results): [3] instead of “(Intergovernmental Panel on Climate Change, 2018)”.

Line 286 (Results): add a briefly description of the workers interviewed using data regarding categories such as gender, age range, studies, occupation (kitchen staff, guides, maintenance team, etc.), workplace (head office, game reserve), etc. This information is basic, and you should have recorded it. If you do not want to increase the length of the article, send it as a supplementary material.

Line 307: do not use a paragraph break here. Merge paragraphs in lines 306-319.

Line 367 (Results): add the reference (Satake, 2019) to the reference list. This change, again, would modify the references numbering.

Line 399 (Discussion of results): [56] instead of [55].

Lines 435-476: Please, review all this text and delete the journal’s guidelines for address these subsections about the paper.

References:

  • Add full stops at the end of lines 493, 509, 511, 524, 526, 530, 532, 541, 559, 565, 572, 574, 576, 586, and 591.
  • Add missing data at lines 485 (“70(1), 8-12.”), 528 (final page number), and 572 (final page number).
  • Do not use capitals for the title in lines 499-500.
  • Do not add a point after IPCC in lines 482, 537, and 594.
  • Line 514, delete a blank space after “Nature”.
  • Line 579, delete a parenthesis and a point after the author information.
  • Lines 580-582, delete the repeated text, the reference is two times.

Author Response

Reviewer response attached 

Reviewer 3 Report

The manuscript ‘Global warming and its implications on nature tourism at Phinda Private Game Reserve, South Africa‘ provides a valuable insight into practical aspects and impact of global warming on nature tourism. It meets the criteria of a scientific paper and contributes to science by its findings based on field research carried out in Phinda Private Game Reserve, South Africa. The theoretical framework is based on a sufficient review of adequate literature, the method is described clearly allowing study replicability, the results are interpreted appropriately and they make a sound basis for drawing conclusions provided by the Authors.

However, section 5 Discussion of results is rather insufficient, while section 4 Results includes elements that clearly belong to discussion section. These are all references to literature, e.g. lines 249-252; 277-285; 308-319, etc. etc. They should be moved to section 5, enhancing it significantly and improving the structure of the manuscript.

Section 4.3. Impact of Global warming on tourism is of major importance for the manuscript as it discusses the implications of global warming on nature tourism, which is the aim of the study. The discussed effects of warming on employees’ performance, on tourists, on the equipment and last but least on animals are of crucial importance. Thus there is a question whether – based on the field research – the section could be extended.

There are also several minor things that should be improved:

  • minor language corrections are required, eg. No full stop after the title (line 3), correct the beginning of the sentence in line 31, correct the sentence in line 53; 78; etc;
  • editorial corrections are necessary e.g. references in lines 257-258 and 367 are different than in the whole text.

Taking into consideration the overall merit of the manuscript I recommend it to be accepted and published, after a major revision of sections 4 and 5.

Author Response

Reviewer response attached 

Reviewer 4 Report

The article "Global warming and its implications on nature tourism at Phinda Private Game Reserve, South Africa" is an interesting and well-written epirical study on the local challanges faced by tourism industry in relation to the global climate change. I am in favour of the publication, yet I will point to some minor issues that should be addressed to improve the quality of the presentation of the research:

  1. Line 150: the word "growth" (of temperature) is missing - it was the growth in temperature that averaged between 0.45 and 0.86 degrees Celcius.
  2. Figure 2 - there is a typo in "Report Production".
  3. Subsection 4.1 title is misleading. The subsection does not include any "Evidence of global warming" (this evidence is clear, but is not a topic of this particular paper). It rather describes the LOCAL climate change. So the title should be "Evidence of local climate change".
  4. Lines 217-219 - the sentence is not clear, and it says about average MINIMUM temperature, which is missing.
  5. Line 220 - strictly speaking you cannot write about "significant alpha of 0.05". 0.05 is a significance level and is denoted as alpha. So it should be rather "lower than the assumed level of significance (alpha) of 0.05.
  6. I would be very careful in estimating statistical significance of change in temperatures in section 4.1. First, the dataset is small and of relatively low precision (up to 0.5 degrees). Second, more importantly, I do not really see the point in calculating statistical significance here. Obviously, there is a discussion on the sense of using significance on population data. In this case you have the population data (all years in the period) which you cannot treat as a sample of any larger dataset both temporarily (in other periods climate had different dynamics) and spatially (in other places the dynamic is different). So I would claim that inclined trend lines drawn on figure 3 are sufficient evidence that there has been a change in the period and no statistical tests are needed to assure it.
  7. Lines 225 and 228 - words "increase" are missing.
  8. Table 1 - "opening" and "closing" temperature captions are unclear - do you mean 1994 and 2019?
  9. Calculating statistical significance of differences in temperatures in specific months is even more problematic than in case of yearly temperatures. I would omit it totally, instead presenting an more informative graph or a table summarising the changes in temperature of specific months. If not, remember that you are making a multiple comparison here and you should make an alpha correction there.
  10. I think the section 4.2 should not be separated from 4.1. It says about the same thing (changes in local climate) and is separated from the part 4.3 by the data it uses (climate data vs interviews). So it would be more clear to merge 4.1 with 4.2.
  11. Lines 265-276 - maybe try to be a bit more concise. The sentences in the paragraph seems to repeat the same information several times, it could be shortened.
  12. Title of the section 4.3 (which could be 4.2, see two notes above) should be more clear: "Impact of global warming on tourism according to interviews".
  13. Lines 343 and 346 - by "massage spar" did you mean "massage spa"?

Author Response

Reviewer response attached 

Reviewer 5 Report

The topic is interesting, to analyze the effect of climate change on a tourist destination, although in principle it seems far from the research topics of the special issue of the journal "Climate Change, Air Pollution, and Human Health".

The introduction does not establish specific objectives, which allow to visualize the relationship between the change in temperature and human health, by focusing on changes in temperature and tourism operations in general, without focusing specifically on "human health", rather it appears as collaterally the field of human health.

In this sense, there is a single research question: how does the increasing temperature in the context of global warming affect nature-based tourism operations in South African private game reserves? And the justification for the novelty of the work is also oriented towards that generic question not towards human health.

There is repetition in some introductory paragraphs and literature review. In the literature review the appropriate literature is not reviewed, so for example, it does not make sense to talk about the effect on Winter Olympics, a review of the literature more specifically oriented to the combination of health, tourism and climate change is recommended. Nor does it seem to dominate the concept of tourism destination that is mistakenly handled.

At the end of the literature it is stated that "The study also examines the impacts of global warming, particularly the increasing temperatures on flora and fauna, at Phinda Private Game Reserve in South Africa. "So it is finally ambiguous is the objective of the work. Nor are hypotheses of the work to be contrasted contemplated.

As for the methodology, it is unclear, A mixed-method approach in principle to answer several questions (which are not stated in the work).

Nor does the work include the interviews that are carried out, the criteria for selecting the participants who provide data, requirements that must be met and why. It is necessary to present the topics that address the interview questions ... etc. The steps of content analysis are not adequately explained to be understood.

A clear structure should also be established in the presentation of interview results, with topics, sub-themes, categories .... ect.

The document must be taken care of, there are typographical errors in words such as in Figure 2 "Report Prodcution".

In summary, and after careful reading of the document, a refocus is recommended, in terms of literature review, in the second part of the article the effect on tourism employees is addressed, an issue not reflected in the title or in literature review. Hypotheses must be incorporated. The methodological part must also be reinforced and clarified, focusing on how its application allows hypotheses to be tested. The results must explain the answers to the research questions with the support of the review literature. Finally, the conclusions need to be deepened.

Author Response

Reviewer response attached 

Round 2

Reviewer 2 Report

A brief summary

The authors have tried to combine the comments from five reviewers, something that I understand is always difficult. From my point of view, the final result is adequate to what was requested by me. However, there are some details that I highlight to be corrected before the final publication.

Specific Comments

Citation: At the first review stage I said: “I do not know why you use names in citation for the two first authors instead their surnames”.  The authors answer was: “It is not clear which section or authors the reviewer is talking about”. I am talking about the “how to cite the paper” section on the left margin of the first page. The current text say: “Citation: Zinzi, S.; Dube, K.; Lekaota, L. Global warming and its implications on nature tourism at Phinda Private Game Reserve, South Africa. Int. J. Environ. Res. Public Health 2022, 19, x. https://doi.org/10.3390/xxxxx”. I do not know if two fist authors are correctly cited here.

Line 16 (Abstract): delete one “activities”, now is two times.

Line 34 (Title): Use capitals for the title, “Background and Introduction”.

Line 42 (Background and Introduction): The verb is absent in this phrase. Maybe “…world fails to drastically REDUCE its carbon emissions…”.

Lines 81-83 (Background and Introduction): The correct reference supporting this text is current #8 (Ruti et al. 2016). Please, revise it.

Line 115 (Background and Introduction): here, you should use “Dube et al. [41], and Dube and Nhamo [42] show that…”.

Line 192 (Background and Introduction): Part of this sentence is missing, specifically the beginning. Please, revise it.

Line 215 (Figure 1, caption): I had problems interpreting the main section of the figure because I did not know what the white area was. García et al. (2015) say: “Light grey polygons show state-run protected areas. The white background is a mosaic of agricultural and communal lands and the light blue polygon shows the Lake St-Lucia.” I think this text is important to easily understand the figure. You can use this same text.

Line 246 (Research Methodology): please, delete the “2” before “The study used a…”.

Lines 294-302 (Results and Discussion): Please, be sure all results are equal after change the analysed period.

Line 306 (Figure 3, caption): “1994-2021” instead of “1995-2020”.

Lines 309-309 (Results and Discussion): This sentence is messy and wrong. “Although there was no statistically significant average annual average maximum temperature”. Maybe, use this one: “Although there was not a statistically significant increase in annual average maximum temperature…”.

Lines 319 and 329 (Tables 1 and 2): I am confusing. Data at the fourth column are from 2019 and 2021, or only from 2021 and there is a typo? Please, check this.

Line 344 (Figure 4): Viewing the figure, I think it shows data from 1997 to 2020. Please, check if you should have updated the figure and you didn't.

Lines 504-505 (Results and Discussion): I think the sentence would be clearer if you add some additional words. “One of the popular activities enjoyed by tourists is the WATCHING OF turtles which use the area for breeding”.

Line 556 (Author Contributions): Please, also delete “The following statements should be used”.

Lines 576 (Acknowledgements): “andBeyond”, instead of “&Byond”.

References: Please, be sure you are following the journal’s guidelines regarding citation. I have not reviewed this section in detail.

Reviewer 3 Report

I recommend the manuscirpt to be published in it present form. Good luck!
